# Fish Models for Exploring Mitochondrial Dysfunction Affecting Neurodegenerative Disorders

**DOI:** 10.3390/ijms24087079

**Published:** 2023-04-11

**Authors:** Takayoshi Otsuka, Hideaki Matsui

**Affiliations:** Department of Neuroscience of Disease, Brain Research Institute, Niigata University, Niigata 951-8585, Japan; totsuka@bri.niigata-u.ac.jp

**Keywords:** mitochondria, neurodegenerative disorders, zebrafish, medaka, turquoise killifish

## Abstract

Neurodegenerative disorders are characterized by the progressive loss of neuronal structure or function, resulting in memory loss and movement disorders. Although the detailed pathogenic mechanism has not been elucidated, it is thought to be related to the loss of mitochondrial function in the process of aging. Animal models that mimic the pathology of a disease are essential for understanding human diseases. In recent years, small fish have become ideal vertebrate models for human disease due to their high genetic and histological homology to humans, ease of in vivo imaging, and ease of genetic manipulation. In this review, we first outline the impact of mitochondrial dysfunction on the progression of neurodegenerative diseases. Then, we highlight the advantages of small fish as model organisms, and present examples of previous studies regarding mitochondria-related neuronal disorders. Lastly, we discuss the applicability of the turquoise killifish, a unique model for aging research, as a model for neurodegenerative diseases. Small fish models are expected to advance our understanding of the mitochondrial function in vivo, the pathogenesis of neurodegenerative diseases, and be important tools for developing therapies to treat diseases.

## 1. Introduction

Neurodegenerative disorders are characterized by the progressive loss of structure or function of neurons and include Alzheimer’s disease (AD), Parkinson’s disease (PD), Huntington’s disease (HD), amyotrophic lateral sclerosis (ALS), spinocerebellar ataxia (SCA), and multiple-system atrophy (MSA). The progressive loss or dysfunction of selectively vulnerable neurons leads to a multitude of symptoms, including memory loss, movement disorders, and behavioral changes [1]. Oxidative stress and inflammation are thought to contribute to progressive neurodegenerative disorders, but the detailed mechanisms of these pathologies remain to be elucidated [2,3,4,5]. In addition, the multifactorial etiology and lack of established biomarkers to predict disease progression contribute to the challenges associated with neurodegenerative disorders. One of the most significant risk factors for neurodegenerative diseases is aging. Although various mechanisms of aging have been proposed, it has long been believed that increased reactive oxygen species (ROS) originating from mitochondria cause oxidative damage, leading to cellular dysfunction and tissue failure (the mitochondrial free radical theory of aging; MFRTA) [6,7,8]. Notably, several studies have shown the link between ROS generation and oxidative stress during aging; however, this theory remains controversial [9].

Mitochondria are unique intracellular organelles that are covered by a double membrane, have their own genome, and can self-replicate independently. They are present in all nucleated cells and perform many functions, including cellular metabolism, energy production, and homeostasis. Mitochondrial damage and dysfunction are caused by mutations in nuclear DNA that encode mitochondrial proteins or mitochondrial DNA (mtDNA), and by cellular stress due to environmental factors. There is a link between pathophysiological changes in several neurodegenerative diseases and mitochondrial dysfunction associated with aging, including oxidative stress and reduced adenosine triphosphate (ATP) production capacity [10,11,12,13,14,15,16]. Notably, the loss of neurons is caused by apoptosis regulated by mitochondria [17]. Therefore, mitochondria could be promising therapeutic targets for preventing age-related diseases.

In this review, we will discuss the roles of mitochondria that affect the process of neurodegenerative disorders. First, we will highlight the connection between mitochondrial dysfunction and neurodegenerative disorders. Then, we will introduce the small fish models (zebrafish, medaka, and turquoise killifish) as beneficial in vivo vertebrate models for studying mitochondrial biology. We will summarize several approaches to studying mitochondrial function using small fish and discuss the advantages and challenges. Additionally, we will discuss the potential of small fish models to contribute to the development of therapeutic strategies for age-related neurodegenerative disorders. It is important to emphasize that mitochondrial dysfunction is not the only factor in aging and neurodegenerative disease, but this review will facilitate understanding of this aspect.

## 2. Mitochondrial Dysfunctions Affecting the Neurodegenerative Process

Mitochondrial functions are diverse and complex and essential for cellular homeostasis and survival. Therefore, mitochondrial dysfunction with age leads to cell death and contributes to the progression of neurodegenerative disorders [17] (Figure 1). Mitochondria are estimated to contain 1000–1500 kinds of proteins, of which only 13 are encoded in mtDNA and the rest are encoded in nuclear DNA [18,19]. Proper mitochondrial function depends on the quality control system, such as the transport and translocation of proteins, the turnover of proteins via the ubiquitin–proteasome system, mitochondrial dynamics, as well as the elimination of mitochondria through mitophagy [20]. In the following subsection, we will describe the age-related decline in mitochondrial function and its relation to neurodegenerative disorders.

### 2.1. DNA Mutations

Mitochondrial dysfunction is often caused by mutations in nuclear DNA involved in mitochondrial components and maintenance or in mitochondrial DNA. DNA is exposed to both exogenous physical, chemical, and biological stress and endogenous stress from the production of ROS and failed DNA replication. Chromosome aneuploidy caused by abnormal mitosis in the aged brain has been implicated in neurodegenerative diseases [21]. Aneuploidy is also known to lead to mitochondrial dysfunction and increased ROS production, as well as an acceleration of cellular senescence [22]. In addition, mitochondrial DNA is more prone to accumulate mutations than nuclear DNA [23,24,25]. This is due to the lack of histones, ROS generation in the inner membrane, limited repair mechanisms in mitochondrial DNA, and higher replication frequency than nuclear DNA [26,27,28,29]. Moreover, mitochondrial DNA has very few non-coding sequences, with the result that mutations affect functional genes directly [29]. It might be challenging to protect DNA from mutations with age; therefore, the maintenance of its quality control system is essential. Proper regulation of the balance between the removal of damaged mitochondria and the biosynthesis of new mitochondria is important for aging and longevity [30,31].

### 2.2. Energy Production

The primary function of mitochondria is to generate ATP via oxidative phosphorylation (OXPHOS). This reaction is carried out by the electron transport chain (ETC) consisting of four respiratory chain (RC) complexes (complexes I-IV) and ATP synthase (complex V), which are present in the mitochondrial inner membrane [32,33,34] (Figure 2). High-energy phosphate production is achieved by coupling electron transfer to proton translocation across the mitochondrial inner membrane, resulting in an electrochemical gradient. It has been suggested that the loss of OXPHOS function may cause various disorders, including non-functional synapses, axonal degeneration, increased ROS production, and cell death [35,36]. Cytochrome c is a small protein tethered to the mitochondrial inner membrane by cardiolipin and functions as an electron donor and receptor during OXPHOS. The release of cytochrome c from mitochondria promotes apoptosis via activation of caspase-9, and the subunit of the RC complex acts as a substrate for caspase [37,38,39,40]. Moreover, age-related decline in ATP levels promotes necrotic inflammation, which may trigger a progression of age-dependent disorders [41].

There is a link between declines in overall bioenergetic function and the phenotype of aging [42,43]. The brain is particularly vulnerable to a decrease in bioenergetic function due to its high energy demands and relatively high mitochondrial mass. With aging, decreased activity of RC complex I, decreased ATP production capacity, and cytochrome c release have been observed in the brain [44,45,46]. An increased number of cytochrome c oxidase (COX)-deficient neurons with age have also been reported in the substantia nigra and hippocampus in normal human brains [47,48]. Several studies have shown mitochondrial dysfunction and reduced mitochondrial complex I activity in the substantia nigra and frontal cortex of PD patients [49,50,51]. Similarly, mitochondrial complex I dysfunction has been reported in the skeletal muscle and platelets of PD patients [52]. In addition, the induction of a familial PD gene mutation into neuronal cells caused defective complex I activity and synaptic function [53]. Reduced complex IV activity was also observed in postmortem homogenates of AD and PD patients [54,55,56]. Interestingly, a decrease in mitochondrial respiration associated with a decline in the electron transfer rate of complexes I and IV among RC complexes was consistently observed in aging and neurodegenerative diseases. On the other hand, it has been reported that decreased complex II and III activity with increased complex I and IV activity occurred in MSA cerebellar white matter [57]. In addition, decreased mRNA expression of all mitochondrial complexes subunits (I-V) has been observed in the frontal cortical and angular gyrus in PD with dementia [58]. The relationship between these disease-, symptom-, or region-specific alterations of RC complex and neurodegenerative pathogenesis needs to be elucidated in future studies.

### 2.3. Reactive Oxygen Species/Oxidative Stress

The mitochondrial RC is the primary site of ROS production in the cell [59,60] (Figure 2). ROS produced in the OXPHOS process oxidizes nucleic acids, lipids, and proteins, causing damage, especially within the source origin, mitochondria [61,62,63,64]. Mitochondria possess antioxidant systems to prevent oxidative damage, and properly regulated ROS can trigger various signaling pathways and regulate autophagy [65,66,67]. However, accumulated oxidative damage to mitochondria due to aging and other factors can affect ATP production and other essential functions in mitochondria [68,69,70]. Moreover, ROS themselves also increase mitochondrial membrane permeability, leading to additional ROS release (ROS-induced ROS release, RIRR) [60,71].

The negative cycle associated with ROS production significantly impacts survival in cells with high energy requirements, such as neurons. In addition, the brain is considered vulnerable to oxidative stress due to its high oxygen consumption, an abundance of oxidizable unsaturated fatty acids, and low expression of some antioxidant enzymes [72,73]. Excessive oxidative stress and oxidative changes in mtDNA have been reported in the postmortem brain of AD patients [74,75,76]. Decreased activity of the alpha-ketoglutarate dehydrogenase complex, which is sensitive to oxidants, is a feature found in AD patients’ brains [77,78]. Oxidative damage to the RC complex I was observed in postmortem brain samples from PD patients, as well as oxidative damage to nucleic acids, lipids, and proteins [79,80,81,82]. ALS-associated antioxidant enzyme superoxide dismutase 1 (*Sod1*) mutant mice showed increased ROS production, decreased expression of NF-E2-related factor 2 (*Nrf2*), a stress response sensor, and early onset of ALS-like pathology [83,84,85]. Since oxidative damage plays a central role in the common pathophysiology of neurodegenerative diseases, reducing the harmful effects of ROS in the brain may be a promising treatment option to slow the progression of neurodegeneration and alleviate associated symptoms.

### 2.4. Calcium and Iron Homeostasis

In addition to energy production, mitochondria are the site of critical metabolic and synthetic processes, including fatty acid oxidation, cholesterol synthesis, glucose synthesis, nucleotide synthesis, calcium homeostasis, iron–sulfur clusters (ISC) synthesis, and heme synthesis [86,87]. Here, we focus on the control of calcium and iron levels.

Mitochondrial regulation of calcium levels has a vital role in signaling molecules associated with cell death and cell survival, as well as maintenance of mitochondrial function [88,89]. Mitochondrial regulation of calcium is particularly important in neurons because calcium functions as a second messenger in neurons [90]. To maintain the cytosolic calcium level, the temporary influx of calcium ions that occurs during synaptic activity is taken up by the endoplasmic reticulum and mitochondria and also released to the extracellular space, which requires a large amount of ATP consumption [91,92]. Thus, decreased ATP production capacity affects calcium homeostasis. High cytosolic calcium levels stimulate various Ca^2+^-dependent catabolic enzymes, such as phospholipases, proteases, and endonucleases, resulting in cell death [93].

In HD patients and mouse models, it has been reported that depolarization at lower calcium loads was caused by mitochondrial calcium abnormalities, which occurred earlier than pathological or behavioral abnormalities [94]. Another study showed that increased cytosolic calcium concentration promoted the degradation of wild-type huntingtin via calcium-dependent proteases, leading to the loss of huntingtin neuroprotective activity [95]. Calcium overload in mitochondria also stimulates ROS generation and releases pro-apoptotic factors such as cytochrome c through the perturbation or rupture of the mitochondrial outer membrane, which triggers calcium-induced cell death [96,97]. To sustain the bioenergetic function of mitochondria, the crosstalk with another calcium storage, the endoplasmic reticulum (ER), is also important. The ε4 allele of apolipoprotein E (APOE4) is considered one of the risk factors of AD. Tambini et al. showed upregulated mitochondria-associated ER membrane (MAM) activity in human fibroblasts or mouse neurons when cultured in APOE4-containing medium, which promotes the transfer of calcium from the ER into the mitochondria [98]. In addition, presenilin (*PSEN*) mutations in familial AD have been associated with the dysregulation of calcium signaling. PSEN1/2 are abundant in the ER membrane and interact with ER calcium channels such as inositol 1,4,5-trisphosphate receptors (IP3R) and ryanodine receptors (RyR) [99,100,101,102]. It has also been shown that both increased ER-mitochondrial contact sites and the expressions of MAM-related proteins, such as IP3R, RyR, and voltage-dependent anion channel (VDAC1), were found in neurons from sporadic and familial AD patients and AD mouse models [103].

Among the vital metals in the mitochondria, iron plays a central role and is essential for the function of the RC complex. ISC, which is synthesized in the mitochondria, is used for OXPHOS, cellular iron homeostasis, pyrimidine/purine metabolism, tricarboxylic acid cycle (TCA cycle), DNA repair, and heme synthesis [104]. Excessive free iron generates oxidative stress, which is a hallmark of age-related diseases. Iron accumulation within the central nervous system (CNS) was found in AD, PD, HD, and ALS [105,106,107,108,109,110,111,112]. Agrawal et al. demonstrated that human HD and mouse model HD brains accumulated mitochondrial iron and showed increased expression of the iron uptake protein mitoferrin 2 and decreases in the ISC synthesis protein frataxin [113]. Intracellular free iron causes lipid peroxidation and hydroxyl-radical generation, resulting in cell death known as ferroptosis [114]. Lipid peroxidation can transmit from ferroptotic cells to neighboring cells, inducing a chain of further ferroptosis [115]. In ALS, ferroptosis but not necroptosis plays a central role in selective motor neuron death [116]. Therefore, the association of ferroptosis with the pathophysiology of neurodegenerative disorders has gained researchers’ attention [117,118,119,120,121].

### 2.5. Mitochondrial Dynamics

Mitochondria are dynamic organelles that change their number, size, and DNA copies according to cellular requirements. It has been reported that the copy number of mitochondrial DNA decreases with age [122,123]. Mitochondrial dynamics refers to two opposing phenomena: fission and fusion [124,125]. Both processes are essential for mitochondrial quality control against stress conditions. The rate of mitochondrial fission and fusion depends on metabolic changes and stress intensity. Mitochondrial fission provides a sufficient number of mitochondria to daughter cells during mitosis. Even in non-dividing cells, fission contributes to quality control by isolating damaged mitochondria and targeting them for removal by mitophagy [125,126]. Inhibition of fission in mouse Purkinje cells resulted in morphological abnormalities associated with excess fusion, oxidative damage accumulation, and loss of respiratory function [127]. Excessive mitochondrial fission is an early event in apoptosis and induces apoptosis via permeabilization of the outer membrane [128,129,130]. Mitochondrial fusion can rescue mitochondria with mutations by allowing them to complement each other or mitigate low-level damage by exchanging proteins and lipids with other mitochondria. Therefore, inhibition of fusion leads to the accumulation of mitochondrial damage, resulting in a wide variety of dysfunctions, including heterogeneity of mitochondrial membrane potential, impaired respiratory chain function, disruption of mtDNA integrity, reduced mitochondrial Ca^2+^ uptake, mitochondrial fragmentation, and apoptosis [131,132,133,134,135].

Disturbances in mitochondrial dynamics have been found to escalate pathogenesis in neurodegenerative disorders [136,137]. Heterogeneous mutations in the mitochondrial fusion gene mitofusin 2 (*MFN2*) cause the neurodegenerative disease Charcot-Marie-Tooth type 2A (CMT2A) [138,139]. Loss of *Mfn2* caused neurodegeneration of Purkinje cells in the cerebellum and dopaminergic neurons [140,141]. In brain tissue from patients with AD and HD, increased expression of fission-related genes such as dynamin related protein 1 (*DRP1*) and fission protein 1 (*FIS1*) and decreased expression of fusion-related genes such as *MFN1*, *MFN2*, and optic atrophy 1 (*OPA1*) have been reported, suggesting that excessive fission inducing apoptosis occurs [142,143]. Abnormal interaction of accumulated amyloid-β with DRP1 accelerated mitochondrial fragmentation in AD [142]. Mutant huntingtin also has been reported to interact with DRP1, increasing its enzyme activity and promoting fission [144,145]. Selective inhibition of DRP1 suppressed excessive mitochondrial fragmentation and improved mitochondrial function in cell models of HD and cells derived from HD patients [146]. These disruptions in mitochondrial dynamics potentially have a significant impact on the process of mitophagy.

### 2.6. Mitophagy

Mitophagy is the removal of dysfunctional mitochondria by autophagy-mediated fusion with lysosomes, which maintain proper cellular homeostasis [147,148]. Mitophagy pathways can occur in response to disturbances such as decreased membrane potential and accumulation of misfolded proteins, and selective mitochondrial fission plays an important role [126,149,150]. Recessive mutations in PTEN-induced putative kinase 1 (*Pink1*) and Parkin (*PARK2*) have been identified as genetic causes of familial PD [151,152]. PINK1 is a mitochondria-localized serine-threonine kinase that can phosphorylate ubiquitin to activate Parkin, and Parkin is an E3 ubiquitin ligase in the cytoplasm, and both play central roles in inducing mitophagy [153,154]. In response to mitochondrial damage, such as loss of mitochondrial membrane potential or accumulation of misfolded proteins, PINK1 stabilized on the mitochondrial outer membrane, and Parkin migrated from the cytosol to the damaged mitochondria [153,155]. Disturbed autophagy systems have been reported in other neurodegenerative disorders, such as AD and HD [156]. In HD cellular models, autophagic vacuoles failed to recognize and trap cytosolic cargo, leading to insufficient autophagy and the accumulation of dysfunctional mitochondria [157]. Mitophagy enhancement inhibited amyloid-β and tau pathology in AD models, suggesting mitophagy could be a potential therapeutic target [158].

### 2.7. Immune System

Mitochondria are thought to have originated as proteobacteria and later became symbiotic in other cells (eukaryotic cells) [159]. Therefore, their components are likely to be recognized as foreign substances by our innate immune system. Mitochondrial DNA is particularly cytotoxic and triggers an innate immune response. In a cultured cell model mimicking Parkinson’s disease, leaked mitochondrial DNA induced an elevated type I interferon response and cell death through the DNA sensor interferon-gamma inducible protein 16 (IFI-16) [160]. In another study, transfection of oxidant-initiated degraded mitochondrial polynucleotides into primary mouse astrocytes stimulated the expression of interleukin 1β (*Il-1b*), *Il-6*, monocyte chemotactic protein 1 (*Mcp1*), and tumor necrosis factor α (*Tnfa*) [161]. In addition to mitochondrial DNA, mitochondrial components such as oxidized cardiolipin, cytochrome c, ATP, N-formyl peptides, and high mobility group box 1 have been reported to induce inflammatory responses [162,163,164,165,166,167,168,169]. Mitochondrial lysates yielded the expression of *Tnfa* and *Il-8* in a mouse microglial cell line [170]. Interestingly, they also upregulated the expression of amyloid-β precursor protein (*App*), a precursor of amyloid-β that accumulates in Alzheimer’s disease brains [170]. The microglia of AD patients express cytokines/chemokines such as TNFA, IL-1B, major histocompatibility complex (MHC) class II, cyclooxygenase 2, and MCP1 [171,172]. Similarly, elevated levels of TNFA, interferon γ, IL-2,4,6, and 10 were found in the serum of PD patients [173]. The release of mitochondrial components associated with cell death may induce an immune response and contribute to the progression of neurodegenerative disease with neuroinflammation.

## 3. Small Fish Models to Study Mitochondrial Function/Dysfunction

Small fish (e.g., zebrafish and medaka) are widely used vertebrate models in developmental genetics and embryology due to the presence of numerous mutants, ease of genetic modification and embryo manipulation, and ease of imaging using transparent embryos and larvae. These have been recognized as human disease models in the last decades because they share a high similarity in genes, organ structures, and disease phenotypes [174,175]. For instance, both zebrafish and medaka have shown PD-like phenotypes by the administration of neurotoxins such as 1-methyl-4-phenyl-1,2,3,6-tetrahydropyridine (MPTP) and 6-hydroxydopamine (6-OHDA) [176,177,178,179,180,181]. In the following subsection, the selected examples of genetic models, imaging techniques, and drug screening illustrate the advantages and challenges of small fish models in studying mitochondrial function/dysfunction (Figure 3).

### 3.1. Genetic Models

Zebrafish and medaka are suitable model organisms to perform gene editing. There are several efficient genome editing methods used for small fishes, such as zinc-finger nucleases (ZFNs), transcription activator-like effector nucleases (TALENs), and CRISPR/Cas9 [182]. Small fish release fertilized eggs outside the body, making it easy to introduce genome editing tools by microinjection. In addition, a morpholino antisense oligonucleotide (MO)-based gene knockdown has been widely performed in small fishes. Currently, the mitochondrial gene dysfunction models are mainly evaluated by MO-based gene knockdown (Table 1). This method is easy to introduce, but the effect is temporal, occurring only during the early developmental stage. Furthermore, it has been reported that many genetic knockout models cannot replicate the MO-induced phenotypes, possibly due to off-target effects [183]. Therefore, it is important to establish knockout models or use spontaneous mutants for analyzing gene function [174]. In addition, tissue-specific promoter and/or site-specific recombinase technology (e.g., Cre-Lox recombination system) are required to study tissue-specific effects. Moreover, an inducible recombination system (e.g., heat-shock promoter, chemical-inducible recombination, and light-inducible recombination) may be necessary to analyze the phenotypes in aged populations or avoid lethality. Detailed strategies for spatiotemporal mutagenesis have been summarized in other reviews [184,185]. Notably, fish have undergone a whole-genome duplication that causes them to possess duplicated genes [186,187]. In most cases, one copy loses its function as a pseudogene (nonfunctionalization). However, other cases involve subfunctionalization in which the two copies split the original function, or neofunctionalization in which one copy generates a new function [188]. It is important to remember this fact when analyzing phenotypes and gene function. Here, we summarize zebrafish and medaka models used to study several genes associated with mitochondrial function and mainly neuronal defects. Other models of neurodegenerative disorders can be found in recent reviews [189,190,191,192].

#### 3.1.1. Neurodegenerative Disease Models

Of the mutated genes that cause familial PD, many encode mitochondria-associated proteins (PINK1, Parkin, PARL, DJ-1, and LRRK2). PINK1 is a protein associated with mitophagy induction through Parkin activation [153]. Knockdown of *pink1* in zebrafish reduced the number of dopaminergic neurons [193,195]. Another study of *pink1* morphants reported no overall decrease in the number of dopaminergic neurons but disturbed patterning and projection of these neurons [194]. Furthermore, the *pink1* null mutant and *pink1* knockout model also showed the loss of dopaminergic neurons [196,197,198]. These results suggest that single depletion of *pink1* in zebrafish is sufficient to affect dopaminergic neurons and a suitable model of PD. PD is also characterized by movement disorders. Motor deficits have also been observed in many of the *pink1* deletion models presented here. Hughes et al. developed a classification method in adult zebrafish movement disorders with PD-like phenotypes using high-resolution video capture and machine learning [198]. These zebrafish models and behavioral assessments will provide further insights into understanding human pathology.

DJ-1 (PARK7) has a role in protecting cells from oxidative and ER stress [226]. Zebrafish knockdown of DJ-1 did not alter the number of dopaminergic neurons; however, they were vulnerable to oxidative stress [201,202]. DJ-1 knockout models showed a reduction in dopaminergic neurons with aging [198,203]. Therefore, mutations in DJ-1 may not directly cause neuronal death, but the weak neuronal cell protection system leads to PD through the accumulation of stress with age.

Leucine-rich repeat kinase 2 (LRRK2) is a multidomain protein interacting with parkin [227,228]. The studies of knockdown or knockout of *lrrk2* have reported various but conflicting phenotypes in terms of the number of dopaminergic neurons [204,205,206,207,208,209]. Notably, the mechanism underlying the pathogenic effect of PD by *LRRK2* mutation remains unknown because point mutations have been found among different domains [228]. The most frequent mutation in *LRRK2* is supposed to be a gain-of-function that increases kinase activity [229,230]. Further investigation will be needed to understand the role of LRRK2 in PD progression by using not only loss-of-function models but also by establishing a gain-of-function model.

Several genetic medaka models of PD have been established. Unlike zebrafish, *pink1* or Parkin (*park2*) single mutations screened from the ENU mutagenesis library did not show dopaminergic cell loss [231,232]. The double deficiency of *pink1* and Parkin (*park2*) led to a deterioration of motor function and loss of dopaminergic neurons [232]. DJ-1 knockout medaka was also established by TALEN and CRISPR/Cas9 systems, but the phenotypes were not reported [233,234]. There have been few analyses of mutants in medaka, and further findings should be obtained in future studies.

Gain-of-function mutations in *SOD1* cause familial ALS. Mutated *SOD1* aggregates in the mitochondrial inner membrane and is thought to be involved in oxidative stress and apoptosis [235]. Lemmens et al. reported abnormal motor neuron branching and short axons in zebrafish, which transiently overexpressed mutated human SOD1 proteins [210]. On the other hand, no such axonal abnormalities were observed in transgenic lines overexpressing mutant zebrafish *sod1*, but abnormal neuromuscular junctions (NMJs) were observed [211]. This line showed end-stage manifestations, including reduced swimming behavior, partial paralysis, reduced number of motor neurons, and mitochondrial vacuolation. Decreased NMJs and motor neurons have also been reported in zebrafish mutants of Sod1 [212]. These models recapitulate the ALS phenotype and can be used as valuable models for ALS research.

#### 3.1.2. Neuronal Defects

Charcot-Marie-Tooth disease (CMT) is a peripheral neuropathy resulting in weaker muscles. Mutations in the mitochondrial fusion gene *MFN2* lead to CMT2A [139]. Zebrafish knockdown of *mfn2* showed abnormal motor neurons and myofiber alignments [213]. In addition, zebrafish *mfn2* mutants showing age-related alteration of NMJ pathology and reduced motile mitochondria have been identified [214]. Both morphants and mutants showed dull motor responses to physical stimuli, making them a good model for CMT2A. Similar abnormal NMJ phenotypes could be found in the knockdown of *slc25a1*, the mitochondrial citrate carrier [217]. Mutations in *SLC25A1* are associated with neuromuscular transmission disorders (congenital myasthenic syndromes) and neurometabolic disorders (D-2- and L-2-hydroxyglutaric aciduria) [217,236].

It has been reported that mutation in the *MFN2* gene impaired mitochondrial axonal transport [237]. Therefore, defective mitochondrial transport along axons may be associated with NMJ pathology and loss of motor function. Zebrafish *kbp* is an ortholog of human Kif1-binding protein (KBP/KIAA1279) that regulates mitochondria localization. The zebrafish *kbp* mutant revealed that *kbp* has an essential role in the development, growth, and maintenance of axons [218]. Notably, mutations in *KIF1B* are associated with CMT2A as well as *MFN2* [238]. Similarly, the zebrafish mutant of *actr10*, part of the dynactin complex, led to mitochondria failing to attach to the dynein retrograde motor, leading to axon swelling and accumulation of mitochondria [219].

#### 3.1.3. Anomaly of Brain Development

Knockdown of mitochondrial genes often leads to systemic effects during embryogenesis. Mitochondrial transcription factor A (TFAM) is a multifunctional protein that regulates the transcription and translation of essential mitochondrial genes, mtDNA copy number, and DNA packaging [239,240]. OPA1 is involved in mitochondrial fusion and regulation of apoptosis, and its mutation is associated with autosomal dominant optic atrophy [132]. SURF1 is a COX assembly protein, and its mutation is associated with Leigh syndrome [241]. Even though these genes possess different mitochondrial functions, the morphants showed severe developmental defects in the eye, heart, and brain regions [223,224,225]. These defects have also been reported in *mfn2* and *slc25a* morphants [213,217]. Mitochondrial gene mutations often cause early-onset mitochondrial diseases such as Leigh syndrome and mitochondrial encephalopathy, lactic acidosis, and stroke-like episodes (MELAS) [242]. Mitochondrial disease is clinically complex and can affect any tissue or organ: encephalopathy, neuropathy, blindness, deafness, myopathy, cardiomyopathy, enteropathy, renal disease, liver failure, and anemia. However, many children affected by mitochondrial disease exhibit tissue/organ-specific symptoms in the early stages of the disease [242]. The fast-developing small fish may be used as a good model to approach the pathogenesis of these mitochondrial diseases. In medaka, the knockdown of holo-cytochrome c-type synthase (*hccs*) showed the phenotype of microphthalmia with linear skin lesions (MLS) through ROS overproduction [243]. Further investigation will be needed by generating tissue- or cell type-specific gene knockout models to understand the tissue/organ specificity and variability of clinical symptoms.

### 3.2. mtDNA Manipulation

The lengths of mitochondrial DNA of zebrafish and humans are 16,596 and 16,569 bp, respectively, and encode 13 protein genes, 22 tRNAs, 2 rRNAs, and a non-coding control region [244,245]. Many mitochondrial DNA variants, such as point mutations and deletions, have been reported as causative genetic defects of various disorders, including PD and AD [246,247,248,249]. Therefore, developing the tools to edit mitochondrial DNA precisely is essential to understand the etiology of mitochondrial diseases. Current major gene editing methods are also applicable to mitochondrial DNA editing: mtZFN [250,251,252], mitoTALLEN [253,254,255,256], and mito-CRISPR/Cas9 [257,258,259,260]. The mito-CRISPR/Cas9 system was also successfully used in the knock-in strategy in the zebrafish model [261]. However, these strategies face difficulty in delivering the editing components into mitochondria [262]. In addition, mitochondria-targeted nucleases selectively reduced mtDNA haplotypes in the germline, eliminating mitochondrial mutations [263,264]. To date, few studies have been successfully established in in vivo models. Recently, Mok et al. engineered a bacterial cytidine deaminase toxin (DddAtox)-based mitochondrial genome editing tool [265]. DddAtox was split into two inactive portions, which were fused with a transcription activator-like effector (TALE) and a uracil glycosylase inhibitor, resulting in DddA-derived cytosine base editors (DdCBEs). DdCBEs were introduced in zebrafish to create a model of mitochondrial disease. This study showed higher efficiency of mitochondrial *nd5* gene mutation associated with Leigh syndrome and MELAS than a mouse model utilizing the same strategy [266,267]. Further new methods will continue to be developed and optimized for precise mitochondrial genome editing for understanding mitochondrial disease and developing therapeutic applications. In this process, small fish can represent strong in vivo models.

### 3.3. Imaging

Imaging mitochondria is useful for monitoring the structural and functional changes during the pathological process, but measuring mitochondrial function in vivo, especially in mammalian models, involves many technical difficulties. Various fluorescent reporters have been developed and used for in vitro live-cell imaging [268]. To understand the mitochondrial dynamics in vivo, a fluorescent protein fused with a mitochondrial localization sequence (e.g., mito-GFP, mito-CFP, and mito-RFP) has been used in mice and zebrafish models [269,270,271,272]. Dukes et al. reported abnormal mitochondrial transport in vivo in a pharmacological PD model using a transgenic zebrafish in which the mitochondria of dopaminergic neurons are labeled with the fluorescent reporter [273]. Recent advances in fluorescent biosensors enable us to observe the behavior of molecules in live cells with high sensitivity. Using pH-sensitive fluorescent protein, Wrighton et al. established a zebrafish model to monitor physiological stress-induced mitophagy [274]. Vicente et al. fused the Ca^2+^-sensitive photoprotein to GFP and established a zebrafish model to monitor both the cytoplasmic and mitochondrial Ca^2+^ during skeletal muscle contraction [275]. A FRET-based ATP biosensor was also used to visualize ATP dynamics in in vivo beating hearts [276]. These recent models will contribute to the elucidation of the disease mechanisms. Since body transparency is only seen during the embryonic and larval stages, intravital imaging within the adult body, such as the brain, is a challenge similar to in mammals. However, there is an option to utilize pigmentation mutants which allow us to see the internal structure in the adult stage to some extent [277].

### 3.4. Drug Screening

Drug screening processes are used to identify compounds of interest. In such processes, zebrafish is a useful model to evaluate toxicity and effectiveness after the in vitro selection. Needless to say, mice or other mammalian models are evolutionarily closer to humans. However, 71% of human genes have at least one zebrafish orthologue [187]. Furthermore, zebrafish provide beneficial features for high throughput drug screening, including small body size, fast development, ease of laboratory management, and production of large numbers of offspring [174]. Zebrafish can be useful models not only for drug screening but also for determining the mechanism of action [278]. Although high throughput drug screening is available for zebrafish, imaging and analysis of a large number of living organisms are still challenging.

As we discussed above, mitochondria have vital roles in cells, and mitochondrial dysfunction contributes to various disorders. Thus, mitochondria are an important drug target for mutations in mitochondrial DNA, mitochondrial component proteins, and restoration of mitochondrial function [279]. A platform of non-invasive and real-time measurements of metabolic changes in zebrafish larvae has been established and used for drug screening of epilepsy [280,281,282]. Zhang et al. conducted drug screening by using *pink1* deficient zebrafish as a model of Parkinson’s disease. Based on a phenotypic screening strategy, they identified trifluoperazine that induces a stress-dependent activation of autophagy to rescue Pink1 deficiency [283]. Another study utilized a nitroreductase-metronidazole system, which induces apoptosis through damage of mitochondrial DNA, to ablate dopaminergic neurons in zebrafish. Through in vivo dopaminergic neuron imaging, the Renin-Angiotensin-Aldosterone System (RAAS) inhibitors were identified as neuroprotective [284]. Zebrafish models for drug screening and disease models will expand more and contribute to future therapeutics.

## 4. Merits and Demerits of CNS Regeneration Capacity in Zebrafish

One of the biggest differences between mammals and zebrafish is the ability of neurogenesis. Zebrafish possess pronounced regeneration capacity in various tissues and organs; therefore, they have been widely used as a model to study complex tissue regeneration [285]. They are able to regenerate their injured CNS, such as spinal cord and telencephalon injuries, with functional recoveries [286,287,288,289,290]. In adult mammals, radial glial cells are recognized as the source of new neurons (neural stem/progenitor cells), which are localized in the restricted regions: the subventricular zone (SVZ) and the dentate gyrus subgranular zone (SGZ) [291,292,293,294] (Figure 4).

Furthermore, radial glial cells in adult zebrafish brains are widely distributed and form neurogenic niches in the telencephalon, diencephalon, mesencephalon, rhombencephalon, and spinal cord [295,296,297,298,299,300] (Figure 4). These progenitor cells were activated following injury and contributed to regeneration [301,302,303]. In addition, another specified stem cell niche has been identified in the zebrafish cerebellum [304,305,306]. Although these features are important for elucidating the molecular mechanisms that can be translated to therapeutic applications for adult mammals, we should keep their endogenous regeneration ability in mind for understanding the pathological process of neurodegenerative disorders. For example, injection of amyloid-β42-derivates in the zebrafish brain could lead to AD-like phenotypes. However, progenitor cells were activated and processed neurogenesis through the Il-4 signaling pathway [307]. Zebrafish also did not exhibit an age-dependent decline in dopaminergic and noradrenergic neurons, which may be supported by their neurogenesis ability [308]. On the other hand, it has also been reported that the number of newborn neurons and oligodendrocytes decreases with age in the zebrafish telencephalon [309]. Further investigations of regenerative capacity in the fish model will provide knowledge addressing the limited neurogenic capabilities in the mammalian brain.

## 5. Turquoise Killifish: A New Model for Neurodegenerative Disorders

In 2003, the turquoise killifish was reported to have the shortest lifespan among vertebrates [310]. Since then, the turquoise killifish has attracted attention as a new small fish model for aging research. It shows remarkable aging phenotypes during its short lifespan of only several months, including organ atrophy, scoliosis, and elevated levels of aging-related acidic β-galactosidase [311,312,313,314]. In the body, aging is accompanied by decreased telomere length, mitochondrial copy number, and antibody production capacity, leading to multiple organ failures [315,316,317]. This fish was used to study the relationship between gene expression patterns in youth and longevity. This study identified that mitochondrial RC complex I genes were less active at a young age in long-lived fish. In addition, partial pharmacological inhibition of complex I by the small molecule rotenone extended its lifespan [318]. There is no doubt that mitochondrial function decreases with age, but further investigation will be needed to develop a strategy for improving mitochondrial function in the aged population.

Declines in neuronal regeneration ability with age have also been reported in the optic nerve and telencephalon [319,320]. Interestingly, brain regeneration in young fish was mainly supported by non-glial neural progenitor cells [320], in spite of the presence of radial glia for neurogenesis [321]. Further characterization of neural progenitor cells in young and aged turquoise killifish is necessary. As for neurodegenerative disorders, neurofibrillary degeneration in aged fish was observed in the optic tectum, telencephalon, and brainstem, as indicated by Fluoro-JadeB staining [322]. The turquoise killifish also showed age-related degeneration of dopaminergic and noradrenergic nerves and progression of alpha-synuclein accumulation, similar to pathological phenotypes observed in human Parkinson’s disease [308]. This feature may help to elucidate the mechanism of solitary Parkinson’s disease, which is not dependent on a genetic component. Another recent study reported the decreased expressions of enzymes, transporters, and receptors of brain serotonin (5-HT) that are related to neurodegenerative/neurodevelopmental disorders [323]. This study also revealed the increased monoamine oxidase (MAO) activity in aged fish. Aging-induced increased MAO activity has also been reported in rodents and human brains [324,325]. MAO is localized at mitochondrial outer membranes, and its elevated activity is thought to be associated with age-related diseases, including neurological disorders via increased ROS production and regulation of bioactive amines such as serotonin and catecholamines [326,327,328]. This emerging small fish model is still in its infancy. It is expected that the extremely rapid aging characteristic will be used to advance our understanding of mitochondrial involvement in disease and the mechanisms of neurodegenerative disorders.

## 6. Conclusions

In this review, we outlined the factors involved in mitochondrial dysfunction in the progression of neurodegenerative disorders and how small fish models can be used to analyze mitochondrial function. Despite many years of research, we do not know much about the mechanisms of neurodegenerative diseases, including how they occur and when they begin. We have not established a therapeutic strategy for their treatment. In addition, whether mitochondrial dysfunction and the progression of neurodegenerative disease are causally associated or correlated is still debatable. The small fish model alone may not be the key tool that unveils everything, and it is important to apply the observed results to other models such as mammals for deeper understanding. However, there are currently various technical difficulties preventing closer investigations, including the analysis of mitochondrial function, which is related to the progression of the disease. Small fish models are undoubtedly useful as vertebrate models for testing new tools that will be developed in the future. Similarly, they are helpful as an entry model for in vivo testing in drug discovery pipelines. These features will facilitate new insights and discoveries to understand human neurological disorders.

## Figures and Tables

**Figure 1 ijms-24-07079-f001:**
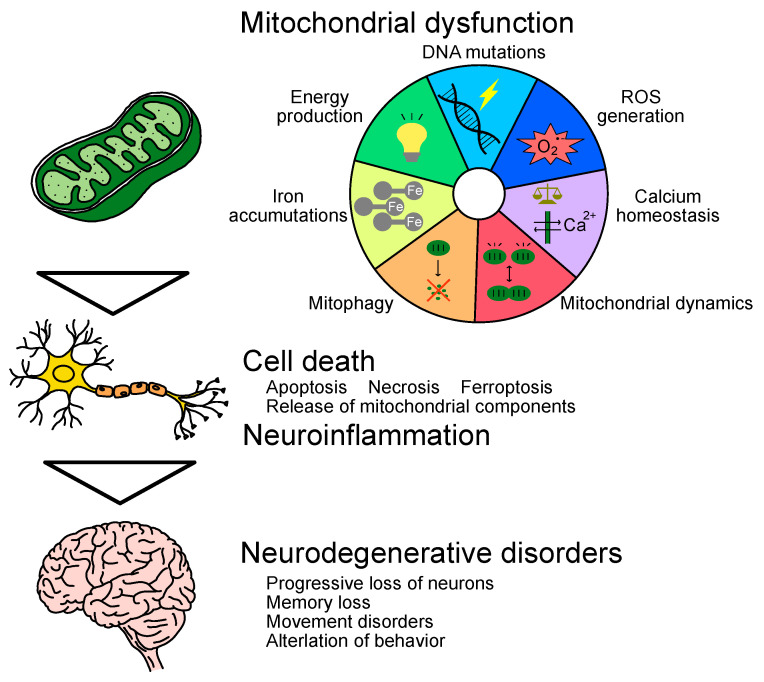
Multifactorial effects of mitochondrial dysfunction in the process of neurodegenerative disorders.

**Figure 2 ijms-24-07079-f002:**
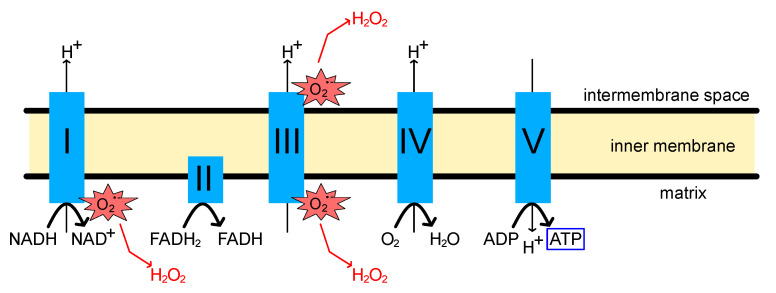
Schematic image of oxidative phosphorylation (OXPHOS) process and reactive oxygen species (ROS) production.

**Figure 3 ijms-24-07079-f003:**
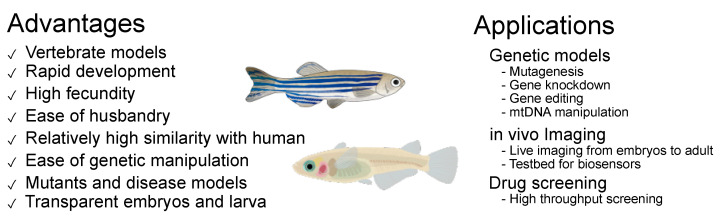
Advantages and applications of small fish models.

**Figure 4 ijms-24-07079-f004:**
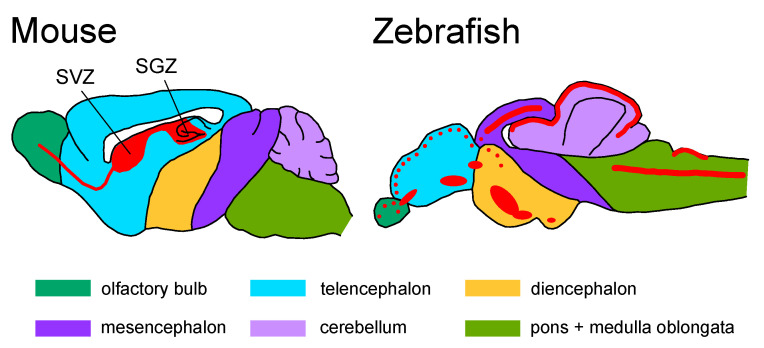
Progenitor cells (radial glial cells) distribution in the mouse and zebrafish brain. The red color indicates regions of constitutive proliferation. Based on the data summary from [295]. SVZ, subventricular zone; SGZ, dentate gyrus subgranular zone.

**Table 1 ijms-24-07079-t001:** The characteristics of loss or gain of function studied in zebrafish for mitochondrial-related genes.

Gene	Related Disease	Model	Phenotypes	Refs
Neurodegenerative disease model				
*pink1* (*park6*)	Parkinson’s disease	MO knockdown	Short tail, small eyes and head. Cardiac edema. Enlarged brain ventricles. Reduced number of dopaminergic neurons. Increased caspase-3 activity and ROS levels.	[193]
		MO knockdown	No alterations in the number of dopaminergic neurons. Disturbed patterning and projection of neurons.	[194]
		MO knockdown	Decreased tyrosine hydroxylase (Th) + neurons.	[195]
		ENU mutagenesis	Reduced number of dopaminergic neurons. Reduced complex I and III activity. Enlarged mitochondria. Increased microglia activity.	[196]
		CRISPR-mediated knockout	Decreased number of dopaminergic neurons and noradrenergic neurons.	[197]
		CRISPR-mediated knockout	Decreased Th + neurons.	[198]
Parkin (*park2*)	Parkinson’s disease	MO knockdown	Decreased Complex I activity. Reduced number of dopaminergic neurons.	[199]
*parl*	Parkinson’s disease	MO knockdown	Increased cell death. Low density or mis-patterned dopaminergic neurons.	[200]
DJ-1 (*park7*)	Parkinson’s disease	MO knockdown	No alterations in the number of dopaminergic neurons. Reduced number of dopaminergic neurons under oxidative stress conditions. Increased Sod1 expression level. Increased apoptosis under proteasome inhibition.	[201]
		MO knockdown	No alterations in the number of dopaminergic neurons. Reduced number of dopaminergic neurons under oxidative stress conditions. Increased apoptosis under the proteasome inhibition condition.	[202]
		CRISPR-mediated knockout	No anomalies in larval development. Small body size. Reduced complex I activity. Reduced Th level in aged fish.	[203]
		CRISPR-mediated knockout	Decreased Th + neurons.	[198]
*lrrk2*	Parkinson’s disease	MO knockdown	Severe embryonic lethality. Small brain, heart edema. Loss of Th + neurons. Deletion of the WD40 domain: Loss of Th + neurons.	[204]
		MO knockdown	No alternation in the number of dopaminergic neurons.	[205]
		MO knockdown	Edema, ocular abnormality, abnormal body axis. Reduced number of dopaminergic neurons. Increased ROS level. Increased Sod1 expression level.	[206]
		ZFN-mediated knockout	A weakened antibacterial response.	[207]
		CRISPR-mediated knockout	Increased apoptosis. Reduced number of microglia/leukocytes in the larval brain. Decreased Th + neurons in the larval brain. Progressive increase in monoamine oxidase-dependent catabolism.	[208]
		CRISPR-mediated knockout	No alterations in the number of dopaminergic neurons.	[209]
*sod1*	Amyotrophic lateral sclerosis	Mutant human *SOD1* overexpression (temporal)	Abnormal axonal branching. Short axonal length.	[210]
		Mutant zebrafish *sod1* overexpression (stable)	No effect on motor axon outgrowth. Abnormal neuromuscular junction (NMJ). Progressive deficiency in locomotion. (end-stage) with intermittent paralysis. Decreased number of motor neurons. Vacuolated mitochondria.	[211]
		ENU mutagenesis	Decreased NMJ and motor neurons.	[212]
Neuronal defect				
*mfn2*	Charcot-Marie-Tooth type 2A	MO knockdown	Irregular somite, small eyes, edema in the brain (mild), and small head with encephalic necrosis (severe). Abnormal axonal projections. Underdeveloped motor neurons. Decreased distribution of AChR clusters. Reduced size of myofibers.	[213]
		ENU mutagenesis	Age-related alteration of NMJ pathology. Reduced number of motile mitochondria.	[214]
*gdap1*		MO knockdown	Reduced density of sensory neurites. Decreased temperature-related activity.	[215]
		MO knockdown	Co-suppression of *mfn2* + *gdap1*: Exacerbated phenotype of motor neuron pathology (failed neuronal extension and innervation of myotome) compared with single suppression.	[216]
*slc25a1*	Congenital myasthenic syndromes/ D-2- and L-2-hydroxyglutaric aciduria	MO knockdown	Abnormal NMJ. Edema of the hindbrain, heart, yolk sac, and tail.	[217]
*kbp*	Goldberg-Shprintzen syndrome	ENU mutagenesis MO knockdown	Delayed development of peripheral axons. Defects in axonal outgrowth. Axonal degeneration or retraction. Abnormal myelination, microtubule organization, and localization of mitochondria.	[218]
*actr10*		ENU mutagenesis TALEN-mediated knockout	Axonal swelling, accumulation of mitochondria.	[219]
PGC-1α (*ppargc1a*)	Wallerian degeneration	Laser axotomy + PGC-1α overexpression	Increased mitochondrial density, attenuated roGFP2 (redox-sensitive sensor) oxidation, delayed degeneration.	[220]
		SNCA (aSyn) overexpression + PGC-1α overexpression	Mediated Snca (aSyn) toxicity in axonal neurons.	[221]
*nipsnap1*		CRISPR-mediated knockout	Reduced mitophagy in the head region. Increased ROS production and apoptosis. Loss of dopaminergic neurons.	[222]
Anomaly of brain development				
*tfam*		MO knockdown	Decreased mtDNA copy number and OXPHOS activity. Edema, small eyes and brain, non-looped heart, disorganized skeletal muscles.	[223]
*opa1*	Optic atrophy	MO knockdown	Abnormal blood circulation, non-looped heart. Small eyes and pectoral fin buds. Obscure midbrain-hindbrain boundary → Enlarged hindbrain ventricle.	[224]
		MO knockdown	Disturbed mitochondrial network. No effect on sensory neurites and temperature-related activity	[215]
*surf1* *cox5aa* *cox5ab*	Leigh syndrome	MO knockdown	Impaired COX activity. Shortened rostral-caudal body axis. Abnormal swim bladder, head shape, gut development, jaw formation. Edema, small eyes, and non-looped heart.	[225]

## Data Availability

The data and tools described in this manuscript are available upon reasonable request.

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
