# Peer review of "Fish Models for Exploring Mitochondrial Dysfunction Affecting Neurodegenerative Disorders"

_ijms, 2023, doi:10.3390/ijms24087079_

Round 1
Reviewer 1 Report
In this paper the authors reviewed accurately and successfully the connection between mitochondrial dysfunction and neurodegenerative disorders and reviewed zebrafish models with neurodegeneration and mitochondrial dysfunction. They only mentioned one model with mitochondrial dysfunction on turquoise killifish, probably because there is not much literature on it.
As the article is called: “Fish models for exploring mitochondrial dysfunction affecting neurodegenerative disorders” and they said that they are going to review small fish models of neurodegeneration and mitochondrial dysfunction it seems plausible to assume that they are going to review medaka models. Consequently, it is surprising that is no mention of medaka through the paper. Isn´t there any model of medaka with neurodegeneration and mitochondrial dysfunction along the literature?
In addition, I strongly recommend checking guidelines for human gene and protein nomenclature and zebrafish gene and protein nomenclature throughout the entire document:
https://www.ncbi.nlm.nih.gov/pmc/articles/PMC7494048/
https://www.ncbi.nlm.nih.gov/genome/doc/internatprot_nomenguide/
https://zfin.atlassian.net/wiki/spaces/general/pages/1818394635/ZFIN+Zebrafish+Nomenclature+Conventions
Author Response
Thank the reviewer #1 for the supportive addvice.
- ......Isn´t there any model of medaka with neurodegeneration and mitochondrial dysfunction along the literature?
Although medaka has been used as a model organism for many years, there are much fewer studies for neurodegeneration and mitochondrial dysfunction than zebrafish (Wang, Cao, Int. J. Mol. Sci. 2021, 22, 10766). However, we agree with your suggestion and should not exclude the examples of medaka studies. We added several studies using medaka in the manuscript (Page 7, Line 304; Page 10, Line 436; Page 11, Line 517).
- ......I strongly recommend checking guidelines for human gene and protein nomenclature and zebrafish gene and protein nomenclature throughout the entire document:......
Thank you for pointing it out. We reviewed thoroughly and corrected gene and protein nomenclature.
Reviewer 2 Report
Although small laboratory fish, zebrafish and turquoise killifish, have some obvious methodical advantages over laboratory rodent, their wide application as objects of translational biomedicine is restricted by mentality of scientists. Indeed many scientists prefer laboratory rodents since thay believe that fish are further from humans in their physiology than rodents. The authors demonstrate the possibility of using fish in translational studies related to mitochondrial dysfunction. Based on extensive theoretical and experimental material, they show that zebrafish are correct model objects in translational research on the fundamental mechanisms of mitochondrial dysfunctions. It is very important that the authors do not try to consider the fish models in the paradigms of rodent models, but directly consider them as a new type of human pathology models. The review is based on a thorough analysis of a large body of literature, is logically structured, well written and easy to read. The review will undoubtedly be useful for specialists whose main objects of study are these laboratory fish and will serve to popularize fish as objects of translational research.
Minor issues
1. The authors write “fishes” instead of “fish” (12, 279).
2. “1000-1500 proteins” (65) should be replaced by “1000-1500 kinds of proteins”.
3. Enzyme monoamine oxidase (MAO) is located on the outer surface of mitochondria and its activity reflects mitochondria function. For example, MAO activity progressively increases during aging (Banerjee, Poddar, Neurosci. Res. 2015, 92, 62–70; Saura et al., Neurobiol. Aging. 1997, 18, 497–507; Kumar, Andersen, Mol. Neurobiol. 2004, 30, 77–89; Nicotra et al., Neuro-toxicology 2004, 25, 155–165; Evsiukova et al., 2023, 24, 3185). The authors should mention these results in their review.
Author Response
Thank the reviewer#2 for the supportive advices.
- The authors write “fishes” instead of “fish” (12, 279).
It was corrected in the text (Page 1, Line 12; Page 7, Line 299).
- “1000-1500 proteins” (65) should be replaced by “1000-1500 kinds of proteins”.
It was corrected in the text (Page 2, Line 66).
- Enzyme monoamine oxidase (MAO) is located on the outer surface of mitochondria and its activity reflects mitochondria function......The authors should mention these results in their review.
Thank you for your suggestion. We mentioned the relation between increased MAO activity and aging in the manuscript (Page 14, Line 660).
Reviewer 3 Report
The manuscript is comprehensive, however, there is a need to have some illustrations that will simplify the bulky writeup
Author Response
Thank the reviewer#3 for the supportive advice.
......there is a need to have some illustrations that will simplify the bulky writeup.
We have added some illustrations (Fig.2 and 3).